# Vitamin D: An Overview of Gene Regulation, Ranging from Metabolism to Genomic Effects

**DOI:** 10.3390/genes14091691

**Published:** 2023-08-25

**Authors:** Giacomo Voltan, Michele Cannito, Michela Ferrarese, Filippo Ceccato, Valentina Camozzi

**Affiliations:** 1Department of Medicine (DIMED), University of Padova, Via Giustiniani 2, 35128 Padova, Italy; giacomo.voltan@aopd.veneto.it (G.V.); michele.cannito@studenti.unipd.it (M.C.); michela.ferrarese.1@studenti.unipd.it (M.F.); valentina.camozzi@aopd.veneto.it (V.C.); 2Endocrinology Unit, Padova University Hospital, Via Ospedale Civile 105, 35128 Padova, Italy

**Keywords:** vitamin D, VDR, vitamin D target gene, RXR

## Abstract

Vitamin D is a pro-hormone characterized by an intricate metabolism and regulation. It is well known for its role in calcium and phosphate metabolism, and in bone health. However, several studies have assessed a huge number of extra-skeletal functions, ranging from cell proliferation in some oncogenic pathways to antioxidant and immunomodulatory functions. Vitamin D exerts its role by binding to VDRs (vitamin D receptors), which are located in many different tissues. Moreover, VDRs are able to bind hundreds of genomic loci, modulating the expression of various primary target genes. Interestingly, plenty of gene polymorphisms regarding VDRs are described, each one carrying a potential influence against gene expression, with relapses in several chronic diseases and metabolic complications. In this review, we provide an overview of the genetic aspects of vitamin D and VDR, emphasizing the gene regulation of vitamin D, and the genetic modulation of VDR target genes. In addition, we briefly summarize the rare genetic disease linked to vitamin D metabolism.

## 1. Introduction

Vitamin D is a fat-soluble vitamin that has been historically known as a molecule. A deficiency in vitamin D might lead to bone diseases, primarily rickets [1]. The discovery of vitamin D dates back to the first half of the 20th century and, despite it still being named as a vitamin, it is well known that it is truly a pro-hormone, with complex endocrine regulation [2]. Indeed, it binds to cytosolic receptors, located mainly in intestinal cells, and osteocytes, but also in several other tissues, such as muscle cells, hematopoietic cells, and the brain. Vitamin D is consequently transported to the cell nucleus, where is able to interact with DNA and modulate the expression of more than 900 genes [3].

The most important effects of vitamin D are on calcium metabolism and bone mineralization; however, it is involved in several physiological and pathological processes, such as cancer, immune modulation, cardiovascular diseases, and metabolic syndrome [4]. Most vitamin D effects are mediated by vitamin D receptors, which are able to regulate a large number of target genes, influencing, consequently, many cellular pathways. Interestingly, VDRs are actually expressed in almost every type of human cell, and they have been found to modulate the transcription of about 3% of human genes [5].

Moreover, there is increasing evidence of the potential role of several VDR polymorphisms in a huge number of diseases [6], such as hypertension, non-alcoholic fatty liver disease, cancer, obesity, and many more [7]. 

The aim of this review is to provide an overview of the genetic aspects related to vitamin D and VDRs, emphasizing the gene regulation of vitamin D, and the genetic modulation of VDR target genes. In addition, we examine the pathogenic role of the most-known VDR polymorphisms, and report a brief summary of the rare genetic diseases linked to vitamin D metabolism.

## 2. Vitamin D Metabolism and Homeostasis

The two main chemical structures of vitamin D are cholecalciferol (vitamin D3) and ergocalciferol (vitamin D2). There are several exogenous ways to obtain vitamin D, including dietary sources, such as oily fish (vitamin D3), mushrooms (vitamin D2), or enriched foods (vitamin D2 and vitamin D3) [3,8]. However, the dietary vitamin D assumption provides only a minor portion of the total daily human intake [9]. The main source of vitamin D is the production in the skin layers, through exposure to the sun’s ultraviolet B rays, especially in the spectral range of 290–320 nm. This is an example of a photochemical process, which does not require any enzymatic involvement, and that leads to the conversion of 7-dehydrocholesterol to pre-vitamin D. Afterwards, pre-vitamin D undergoes an isomerization to vitamin D, through a thermosensitive non-catalytic process [10]. However, vitamin D is biologically inactive, as it requires further hydroxylation steps to turn into its active form, which is able to activate vitamin D receptors (VDRs) [11]. Hence, vitamin D is transported to the liver, carried in the bloodstream by vitamin-D-binding protein (VDBP), where it is hydroxylated to 25-hydroxylated vitamin D (25(OH)D) [12]. The responsible enzyme is CYP2R1, located in the liver endoplasmic reticulum, which can 25-hydroxylate either vitamin D2 or vitamin D3 [13]. Interestingly, CYP27A1 displays a similar enzymatic activity, but is distributed throughout the whole body, and is not able to 25-hydroxylate vitamin D2 [14]. Other enzymes exerting a 25-hydroxylase action, especially in terms of extrahepatic vitamin D production, are CYP3A4, CYP2J3, and CYP2J2. Anyway, CYP2R1 is undoubtfully the major player [15]. The measurement of the circulating levels of 25 (OH)D is considered the best marker for assessing vitamin D status [14].

The hormonally active form of vitamin D is derived from the additional hydroxylation of a C1-carbon atom, in the proximal renal tubule, leading to the production of 1,25-hydroxylated vitamin D (1,25(OH)D). CYP27B1 is responsible for this metabolic step; although the major expression is predominant in the kidney, it has also been found in other sites, including the placenta, monocytes, and macrophages [16,17]. Interestingly, the extra-renal production of 1,25(OH)D is not dependent on parathyroid hormone (PTH) action; thus, the serum availability and sufficiency of 25(OH)D are the limiting factors for the extrarenal synthesis of calcitriol [18].

The importance of this metabolic step was demonstrated by Kitanaka et al., who reported that patients carrying inactivating mutations of the CYP27B1 gene were characterized by vitamin D-dependent type-1 rickets [19].

Interestingly, the kidney is also the main site at which vitamin D catabolism takes place. Indeed, CYP24A1 is a mitochondrial enzyme that can produce 24,25-hydroxylated vitamin D (24,25(OH)D), an inactive metabolite. Thus, CYP24A1 limits the total amount of 1,25(OH)D in tissues, by accelerating its catabolism, and reducing the pool of 25(OH)D available for 1-hydroxylation. 

These enzymes belong to the cytochrome P450 class, a superfamily of monooxygenase-containing heme groups. Their nomenclature derives from their specific spectral properties. Moreover, they are detectable in a large number of organisms, from bacteria to humans, configuring themselves as a wide and heterogeneous enzyme family. The main feature of p450s is that they catalyze the selective oxidation of many molecules, ranging from the biosynthesis of natural products to the degradation of xenobiotic compounds [20].

The production of 1,25(OH)D is finely regulated through an intriguing series of negative and positive feedback. 1-hydroxylation is primarily enhanced by PTH, via the stimulation of CYP27B1 transcription. Therefore, a low calcium and phosphate intake, and hypocalcemia, with the consequent rise of PTH, result in active vitamin D production [21]. Conversely, the increased levels of 1,25(OH)D suppress both PTH secretion and CYP27B1 activity. Vitamin D catabolism is, instead, mutually regulated, as 24,25(OH)D production is stimulated by 1,25(OH)D itself, and is inhibited by hypocalcemia and PTH [22]. This negative feedback results in a protective mechanism against hypercalcemia.

Fibroblast growth factor 23 (FGF23) is a significant regulator of vitamin D homeostasis, too. Indeed, 1,25(OH)D enhances FGF23 production in bones, which, vice versa, suppresses the expression of CYP27B1, and increases 24,25(OH) production in the kidneys. As a result, the final effect of FGF23 is to reduce 1,25(OH)D secretion, further leading to a consequent decrease in FGF23, too [23,24].

Notably, there are some genetic issues influencing vitamin D homeostasis, as well. For example, Thacher et al. described a group of mutations affecting the expression, or the function, of CYP2R1 that were found with a higher prevalence in patients with rickets [25]. Other mutations were, instead, associated with lower circulation 25(OH)D levels, and a decreased sensitivity to vitamin D supplementation [26].

Interestingly, many single-nucleotide variations (SNVs) in CYP2R1were found to be related to some chronic diseases, such as obesity and asthma, but also to cancer and all-cause mortality [27,28]. However, further investigations are necessary to assess whether the connection between CYP2R1 activity and these chronic diseases is significant.

## 3. Genomic and Non-Genomic Effects of Vitamin D

Vitamin D actions might be considered in two distinct ways: genomic and non-genomic effects. The most important non-genomic effect is probably the enhanced calcium and phosphate uptake from the small intestine [29]. This action is the effector mode of PTH calcium reabsorption, which also occurs through the induction of the synthesis of calbindin, a protein that binds calcium ions and transports them from the lumen to the cytoplasm of gut cells. Interestingly, 1,25(OH)D is able to facilitate the passive absorption of calcium, by increasing the permeability of intercellular tight junctions [30]. In addition, vitamin D can also promote phosphate reabsorption in renal tubules [31]. Calcium and phosphorus are essential to hydroxyapatite formation and, by increasing their intake, vitamin D acts indirectly on the bone [32].

The discussion about the genomic aspect is strictly linked to the vitamin D receptor, as it plays a key role. The VDR is a member of the superfamily of nuclear hormone receptors, which might be considered to be ligand-induced transcription factors. VDRs are expressed in the skin, parathyroid glands, adipocytes, small intestine, colon, and other tissues [33,34]. After binding to 1,25(OH)D, the VDR forms a heterodimer with the retinoid acid receptor (RXR) that translocates to the cell nucleus, joining the vitamin D response element (*VDRE*). VDR/RXR complex is considered the major active transcription unit in regulating the vitamin D target genes’ transcription [35]. Moreover, the *VDRE* depends specifically on the cell type, differing from cell to cell. This might be one mechanism of the action specificity of vitamin D [36]. Surprisingly, the complex 1,25(OH)D/VDR can directly interact the with cAMP-response element-binding protein (CREB), hampering its binding to CRE (the cAMP-response element). These activities seem not to require the presence of liganded VDR heterodimerization with RXR [37]. The main steps of vitamin D metabolism, and its genomic and non-genomic effects, are depicted in Figure 1.

## 4. Genetic Factors Influencing Vitamin D Status

An individual’s vitamin D status depends strongly on environmental factors related to geographical region and lifestyle. However, there are also genetic factors which could influence individual serum vitamin D levels and, consequently, be linked to the pathogenesis of many chronic diseases, such as osteoporosis, cancers, and autoimmune diseases [38,39,40]. Interestingly, an association between epigenetic modifications and vitamin D levels has been identified, as well.

Many studies involving twins and close relatives evaluated the inheritance of hypovitaminosis D, which is estimated to be between 23 and 80%, depending also on the study design and environmental variables [41,42].

Numerous candidate gene studies and genome-wide association studies have been carried out over the years, identifying a series of genetic mutations and polymorphisms affecting the genes encoding the molecules involved in the production and activation of vitamin D, transport proteins, VDRs, and coactivating proteins, and alterations affecting proteins secondarily involved in the regulation of vitamin D expression (e.g., mechanisms related to calcium or PTH concentrations). The most investigated genes are *DHCR7*, *CYP2R1*, *CYP27B1*, *GC*, *VDR*, *CYP24A1*, and *RXR* [43].

### 4.1. DHCR7 (7-Dehydrocholesterol Reductase)

The *DHCR7* gene is mapped on chromosome 11, and encodes for a reductase that is responsible for the epidermal conversion of 7-DHC into cholesterol. Mutations in this gene result in an accumulation of 7-DHC, which leads to Smith–Lemli–Opitz syndrome [43].

Some studies identified the following SNVs, which are localized in the 5’ edge region, and are associated with vitamin D deficiency: rs11234027, rs1790349, and rs12785878 [27]. On the other hand, Zhang et al. found that the SNVs rs1790349, rs7122671, rs1790329, rs11606033, rs2276360, rs1629220, and rs2282618 would be genetic protective factors against hypovitaminosis D [44].

### 4.2. CYP2R1 (Vitamin D 25-Hydroxylase)

The *CYP2R1* gene is located on chromosome 11, and encodes for the main 25-hydroxylase that converts cholecalciferol to 25(OH)D [15].

The rs10741657 polymorphism, which is found in the 5′ edge region, is associated with reduced levels of 25(OH)D. Other SNVs that potentially influence an individual’s vitamin D status are rs12794714, rs10766197 [28], rs1562902, rs7116978 [45], rs2060793, rs1993116 [27], rs11023332, and rs1007392 [46].

### 4.3. CYP27B1 (25(OH)D-1-α Hydroxylase)

The *CYP27B1* gene is located on chromosome 12, and encodes for the most important 1α-hydroxylase, which converts 25(OH)D to the active form 1,25(OH)D [43]. Many genetic variants related to vitamin D expression are described. The SNV rs10877012, situated in the promoter region of the *CYP27B1* gene, is associated with reduced serum levels in 25(OH)D [47]. Two other SNVs that might be associated with hypovitaminosis D are rs4646536, located at the 5′ edge region, and the intronic SNV rs703842 [28].

Hence, the inactivating mutations in the *CYP27B1* gene could lead to a deficient conversion of calcidiol to calcitriol, and are better known as vitamin-D-dependent rickets type 1A (VDDR1A) [48]. Less commonly, some variants affect the *CYP2R1* (VDDR1B) result in a deficient 25-hydroxylation process, hampering the conversion of cholecalciferol to calcidiol. This variant is also known as vitamin-D-dependent rickets type 1B (VDDR1B). These disorders cause impaired intestinal absorption of calcium and phosphate, further leading to hypocalcemia and abnormal bone mineralization [49].

The clinical features vary, depending on the severity of the disease. Patients present soon after birth with rickets, and signs of hypocalcemia, tetany, or convulsions [50].

### 4.4. CYP24A1 (Vitamin D 24-Hydroxylase)

The *CYP24A1* gene is located on chromosome 20, and is a mitochondrial enzyme expressed in several target cells containing VDRs, which catalyzes 25(OH)D and 1,25(OH)D catabolism, as aforementioned. This enzyme prevents the accumulation of toxic levels of these molecules; on the other hand, it prolongs the half-life of 25(OH)D when vitamin D levels are reduced [51]. Interestingly, an intronic SNV (rs17219315) was found to be associated with serum 25(OH)D concentrations [52]. Moreover, Barry et al. described that SNVs such as rs2209314, rs2762939, and rs6013897 might potentially modify the efficacy of cholecalciferol supplementation in increasing 25(OH)D serum levels [45].

### 4.5. GC (Vitamin D Binding Protein)

*GC* is mapped on chromosome 4, and encodes for VDBP, which carries vitamin D to various sites of action, facilitating its activity, as well [53]. The DNA sequence analysis of this gene showed two SNVs at exon 11, causing, respectively, a Glu/Asp amino acid change (rs7041) and a Thr/Lys amino acid change (rs4588), associated with a reduction in vitamin D [54]. Some studies identified many polymorphisms associated with a reduced expression of vitamin D and, also, a higher affinity of VDPB to vitamin D [39]: these included some intronic SNVs, such as rs222020 [28], rs2282679, rs1155563 [27], rs2298849, and rs222035 [55]. Other examples of polymorphisms related to vitamin D status are rs16846876, rs17467825, rs842999, and rs12512631, mapped on the 3′ edge region of the GC gene [56].

### 4.6. VDR (Vitamin D Receptor)

The *VDR* gene encodes for the vitamin D receptor, located on chromosome 12, and contains six promoter regions, and eight exons 2 to 9. The DNA-binding domain (exons 2–4) interacts with the VDRE in target genes, whereas the ligand-binding domain (exons 6–9) binds 1,25(OH)D [39]. Several studies have identified hundreds of polymorphisms in the vitamin D receptor gene, but the functional implication is still largely unknown. The most characterized polymorphic sites in the VDR gene are recognized by the restriction endonuclease enzymes *TaqI*, *BsmI*, *ApaI*, and *FokI*, after which they are named. These polymorphisms are strictly correlated with various diseases, but also with homeostatic processes, such as bone mineralization and calcium imbalance [57]. *FokI* (rs10735810, also known as rs2228570) is located in exon 2, and consists of a C > T nucleotide substitution: the T nucleotide is also referred to as allele “f”, while the C nucleotide is defined as allele “F”. The presence of site “F” results in a three-amino-acid-shortened protein, which is characterized by increased transcriptional activity. *FokI*, in particular, the F-allele, as well as having functional consequences on the structure of the vitamin D receptor, is associated with lower serum 25(OH)D levels [28,58]. *TaqI* (rs731236) is located in exon 9 of the VDR gene, and consists of T > C substitution. The T nucleotide is defined as allele “T”, while the C nucleotide corresponds to allele “t”. This polymorphism occurs in a CpG island, resulting in an influence on the methylation status. *BsmI* (rs1544410) is located in intron 8 of the gene, and consists of an A > G nucleotide substitution; the A nucleotide corresponds to allele B, and the G nucleotide corresponds to allele b. This polymorphism influences transcript stability; moreover, it was found to influence the variation in the UVB-induced 25(OH)D increase, interfering in the interaction with RXRA and CYP24A1, as well [59]. *ApaI* (rs7975232) is also located in intron 8, and consists of a C > A substitution (the C nucleotide is referred to as allele “A”, while the A nucleotide is defined as allele “a”). The functional impact of this polymorphism is not clearly explained. There are two lesser-known polymorphisms in the promoter region of the VDR gene: *Cdx2* (rs11568820) [60] and *GATA* (rs4516035) [61], which are located upstream and downstream of exon 1, respectively, causing a decreased promoter activity in the receptor [6,7].

Even if, currently, the functional significance and the clinical implications of these polymorphisms need to be clarified, every mutation that leads to a decrease in VDR functionality thereby prevents calcitriol’s action. This leads to an impaired intestinal absorption of calcium and phosphate, and is classified as vitamin-D-dependent rickets type II, or hereditary vitamin-D-resistant rickets. The main clinical features comprise progressive rickets disease, which starts to manifest during the first years of life. Total body alopecia is present in severe forms of the disease. In some cases, skin lesions or epidermal cysts can be observed, along with alopecia. The disease presents a broad clinical picture that largely depends on the genotype [50,62].

Interestingly, some studies in humans have demonstrated that low circulating levels of 25(OH)D seem to be associated with a higher plasma renin activity, and higher angiotensin II concentrations [63,64]. In addition, a supplementation therapy with cholecalciferol might reduce the increased renin–angiotensin–aldosterone system (RAAS) activity secondary to vitamin D deficiency [65].

The complex vitamin D/VDR, indeed, is able to bind CRE, consequently preventing the binding of cAMP. The fact that cAMP is one of the main stimulating factors for renin production in renal juxtaglomerular cells makes it easy to understand how vitamin D might play a potential role in hampering the development of arterial hypertension [66]. Notably, in patients affected by arterial hypertension, renin activity was found to be inversely related to 1,25(OH)D levels [67].

Several studies have investigated the *FokI* polymorphism of VDRs as a possible condition linked with a higher risk of arterial hypertension. As aforementioned, *FokI* results in the formation of a truncated protein, which is thought to be associated with an increased production of renin and angiotensin II, thereby promoting the development of hypertension [68,69,70]. Other evidence suggests a further association between *BsmI* and arterial hypertension, as well, especially characterizing male patients [71,72,73].

Interestingly, a possible influence of *BsmI* and *FokI* on the development of, and *BsmI*, *TaqI*, and *ApaI* on the progression of, non-alcoholic fatty liver disease (NAFLD) has been hypothesized, as recently reported [74]. The main polymorphisms affecting VDR are resumed in Figure 2.

Other genes indirectly involved in the control of vitamin D homeostasis and expression have also been studied, despite the evidence being less overt. For example, intracellular domain polymorphisms of the calcium-sensing receptor (CaSR), and polymorphisms of cubilin were studied; however, no associations with vitamin D homeostasis were observed [56,57,58,59,60,61,62,63,64,65,66,67,68,69,70,71,72,73,74,75]. Interestingly, a study investigated the impact of *FGF23* gene variation on phosphate homeostasis and bone health, detecting nine FGF23 polymorphisms, three of which were quite common: rs3832879, rs7955866, and rs11063112 [76]. A study showed significant correlations between the RXR SNVs rs3132299 or rs9409929 and 1,25(OH)D, as well as between rs877954 and 25(OH)D levels [52,53,54,55,56,57,58,59,60,61,62,63,64,65,66,67,68,69,70,71,72,73,74,75,76,77].

### 4.7. Epigenetic Factors Influencing Vitamin D Status

Vitamin D might exert an epigenetic effect in the transcription of several target genes; similarly, vitamin D levels and bioavailability are influenced by epigenetic factors. This is, undoubtfully, a developing field of research. Many studies have suggested the role either of the epigenetic modulation of genes involved in vitamin D metabolism and several pathologies, or the association between the epigenetic modifications of genes involved in vitamin D metabolism and vitamin D status [78,79]. The most common epigenetic mechanisms are the acetylation, methylation, and phosphorylation of histone proteins [18]. Among them, the main one comprises methylation via CpG islands located at a gene’s promoter region, resulting in a lower gene expression. These mechanisms would be responsible for nearly 18% of the vitamin D level variance among the population, as well as being a contributing factor to vitamin D deficiency. 

For example, a few studies have shown that the *CYP2R1* methylation status regulates the effect of calcium and vitamin D intake or radiance on vitamin D serum levels: subjects presenting sufficient vitamin D levels, or taking vitamin D supplementation, show lower methylation at the CpG site of the *CYP2R1* gene [80,81]. Similarly, several studies have shown a correlation between vitamin D status and the methylation levels of the *CYP24A1*, *CYP27B1*, *GC*, *RXRA* and *VDR*, and *DHCR7* genes [82,83,84]. Moreover, some studies have reported a potential role of serum B12 and folate in the regulation of the methylation of *CYP27B1* and *VDR*, respectively [85].

Interestingly, many groups of people might be characterized by different effects of vitamin D supplementation on biochemical vitamin D parameters, epigenetic modifications, and the response of transcriptome-wide vitamin D target genes. Moreover, groups of low, medium, and high responders to vitamin D supplementation have been identified, with different molecular responses in 25(OH)D serum levels [85]. These findings have paved the way for the concept of various responses to vitamin D supplementation that lead to a personalized need for daily intake of vitamin D.

## 5. Genetic Effects of Vitamin D

It is now a well-established opinion that vitamin D exerts numerous extra-skeletal effects. Indeed, vitamin D deficiency is associated with an increased risk for many diseases, thus suggesting its crucial role [4].

These effects range from the modulation of cell growth and differentiation, potentially promoting the carcinogenesis process, to the regulation of immune and muscle function. Moreover, vascular and metabolic actions are described, as well [86,87,88,89].

### 5.1. Genomic Action

The vitamin D receptor is one of the nuclear receptors for steroid hormones that functions as a ligand-activated transcription factor, thereby regulating gene expression. It plays an essential role in the genomic mechanism of action of vitamin D [90,91]. Notably, 1,25(OH)D is one of the most potent regulators of its function [92]. Interestingly, the inner surface of the VDR is characterized by a ligand-binding pocket that is able to enclose and bind 1,25(OH)D with an affinity of 0.1 nM, which is a very high affinity, especially in comparison with other nuclear receptors [93].

Through its activation of the VDR, 1,25(OH)D has direct effects on the epigenome, and versus the expression of more than 1000 genes in several human tissues and cell types, resulting in changes in the transcriptome and proteome, as well. Interestingly, VDR is the unique target of 1,25(OH)D in the cell nucleus. There is still no general descriptive model of the regulatory mechanism of vitamin D target genes. After the binding of 1,25(OH)D, VDR interacts with many other nuclear receptors, forming a multi-protein complex that attaches preferentially to DR3-type binding sites within enhancer regions. This VDR multi-complex contains co-receptors, such as RXR, pioneer factors (*PU.1*, *CEBPα*, *GABPα*, *ETS1*, *RUNX2*, *BACH2*), chromatin modifiers (*KDM1A*, *KDM6B*), chromatin remodelers (*BRD7*, *BRD9*), co-activators (*MED1*), and co-repressors (*NCOR1*, *COPS2*). A mediator complex connects the activated VDR complex with the RNA polymerase II located on a specific gene transcription start site (TSS) [35,93,94,95,96], as shown in Figure 3.

However, these VDR complexes, necessarily, need to interact with their respective genomic-binding sites; hence, they must have access to the open chromatin, named euchromatin [97].

Thus, 1,25(OH)D modulates the epigenome in its target tissue in different ways, involving the chromatin status. Essentially, it could act through direct interaction with chromatin-modifying enzymes, as well as through up- or down-regulating the genes encoding for chromatin modifiers [98]. For example, 1,25(OH)D affects histone markers for active chromatin, such asH3K27ac (acetylated histone H3 at lysine 27), and for TSS regions, such as H3K4me3 (tri-methylated histone H3 at lysine 4) [99]. Moreover, the binding of chromatin-organizing protein to several genes is modulated by 1,25(OH)D, and the organization of various genomic loops of DNAs is vitamin-D-dependent. Thus, 1,25(OH)D is able to affect the three-dimensional structure of chromatin [100].

All these mediated mechanisms, in terms of net effects, are finalized, to increase or decrease the activity of RNA polymerase II, and the mRNA expression of 1,25(OH)D target genes.

Notably, the enhancers of the VDR-encoding gene are located near or far, i.e., promoter-proximally or promoter-distally, and many enhancers are also located in clusters hundreds of kilobases away from their target genes, meaning that the intervening genomic DNA forms a regulatory loop. In this way, the expression of the vitamin D target genes is either increased or decreased [35]. 

To downregulate a gene, the mechanism most often employed is to block one or more of its upregulatory factors. This implies that most of the downregulated target genes must be classified as indirect targets, meaning that vitamin D does not directly downregulate them but, rather, counteracts the upregulation of their expression [101,102].

### 5.2. Immune System Regulation

Vitamin D is able to modulate the expression of the genes involved in innate and adaptative immune functions.

In this contest, both 25(OH)D and 1,25(OH)D act in multiple ways, and in several immune cells, such as macrophages, monocytes, and B- and T-type lymphocytes [103].

In macrophages, the expression of *CYP27B1* is induced via immune-specific inputs, leading to the local production of hormonal 1,25(OH)D at the sites of infection, which, in turn, directly induces the expression of genes encoding antimicrobial peptides.

In this scenario, numerous inflammatory or proinflammatory cytokines are modulated via vitamin D, with the aim of limiting inflammation. Indeed, 1,25(OH)D, through autocrine mechanisms, increases the expression of cathelicidin, which, in turn, exerts antiviral and antibacterial effects. Moreover, 1,25(OH)D acts in a paracrine manner, stimulating adjacent macrophages. Therefore, 1,25(OH)D is able to maintain immune tolerance in APC cells, and finely manage the surface expression of MHC class II molecules, immunogenic cytokines, and co-stimulation molecules [104,105].

IL-10 expression, which is characterized by anti-inflammatory activity, is increased; contrarywise, IL-6 and IL-17, which provide pro-inflammatory and atherogenic effects, are reduced [106].

In addition, 1,25(OH)D seems able to activate and modulate natural killer cells, interfering with their metabolism and immunogenic activity [107].

### 5.3. Focus on Clinical Outcomes

It might be speculated that vitamin D could carry antitumor effects, both directly, by controlling the differentiation, proliferation, and apoptosis of neoplastic cells, and indirectly, by regulating the immune cells that belong to the microenvironment of malignant tumors [108].

Data, mainly obtained from observational studies, show that the levels of circulating 25(OH)D concentration are inversely correlated with the risk of breast, prostate, and colorectal cancers, but not the overall cancer risk [109,110,111,112]. Epidemiologic studies are still inconclusive in determining the true effect of vitamin D on reducing cancer risk, and improving patient outcomes.

Interestingly, supplementation with vitamin D did not result in a lower incidence of invasive cancer [7,113,114].

Moving to another topic, considering the role of vitamin D in the downregulation of the adaptive immune system, it is conceivable that observational studies would find an association between vitamin D deficiency and multiple sclerosis, inflammatory bowel disease, and type 1 diabetes [115,116,117].

The evidence from randomized controlled trials (RCTs) of the effect of vitamin D supplementation on clinical outcomes is inconclusive. Hence, the observation that treatment with Vitamin D might reduce the risk of these diseases is still lacking.

### 5.4. Viral Infections and COVID-19

More recently, a role of vitamin D in the course of the COVID-19 pandemic has been postulated.

Unfortunately, in this context, the data obtained from observational and interventional studies are discordant, too.

Circulating low serum vitamin D levels seem to correlate with the risk and the severity of COVID-19 infection, as well as with high mortality and morbidity. However, the results are unreliable, due to the evidence of many confounding factors, also depending on individual patients’ situations [118,119,120,121].

Currently, definitive conclusions about the utility of vitamin D treatment in the context of the prevention of the infection are conflicting. In addition, reliable interventional data on vitamin D supplementation in hospitalized COVID-19 patients are still lacking [122,123,124].

## 6. Conclusions

Despite vitamin D originally having been discovered through its fundamental role in calcium homeostasis and bone formation, nowadays, vitamin D metabolism and signaling are extensively being studied for also having a critical role in extra-skeletal terms. In this context, genetic alterations affecting vitamin D metabolism might be crucial. Indeed, as reported in this review, there are several genes that, if altered, might lead to dramatic variations in 25(OH)D status, with clinical consequences that may become cumbersome, as shown in vitamin D-dependent rickets. An important aspect to keep in mind is the potential role of vitamin D as a modulator in the fields of carcinogenesis, the inflammatory response, and autoimmune diseases. However, a direct link between a potential target therapy via vitamin D supplementation is still unavailable.

Notably, in the future, new research into the role of gene polymorphism and epigenetic modifications in vitamin D status might open up new methods for the clinical application of a personalized approach. Genetic alterations, indeed, might allow physicians to identify patients who are low, medium, or high responders to vitamin D and, consequently, those who most need vitamin D supplementation.

## Figures and Tables

**Figure 1 genes-14-01691-f001:**
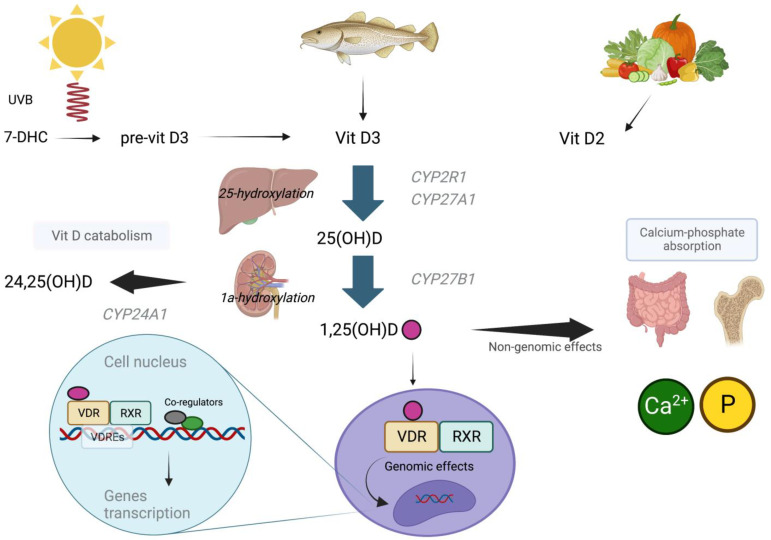
Metabolic pathways regarding the production, activation, and effects of vitamin D. The highest amount of vitamin D is produced in the skin, via the conversion of 7-DHC to vitamin D3. Vitamin D2 is mostly assumed through foods. Vitamin D2 and Vitamin D3 are further hydroxylated in carbonium 25 and 1. 1,25(OH)D exerts its genomic effects by binding to VDR and RXR, and translocating to the nucleus, where it interacts with VDRE. Non-genomic effects are mainly involved in calcium–phosphate homeostasis. 7-DHC: 7dehydrocolesterol, vitamin D3: vitamin D3, vitamin D2: vitamin D2, 25(OH)D: 25-hydroxylated vitamin D, 1,25(OH)D: 1,25-hydroxylated vitamin D, VDR: vitamin D receptor, RXR: retinoic acid receptor, VDRE: vitamin D response element.

**Figure 2 genes-14-01691-f002:**
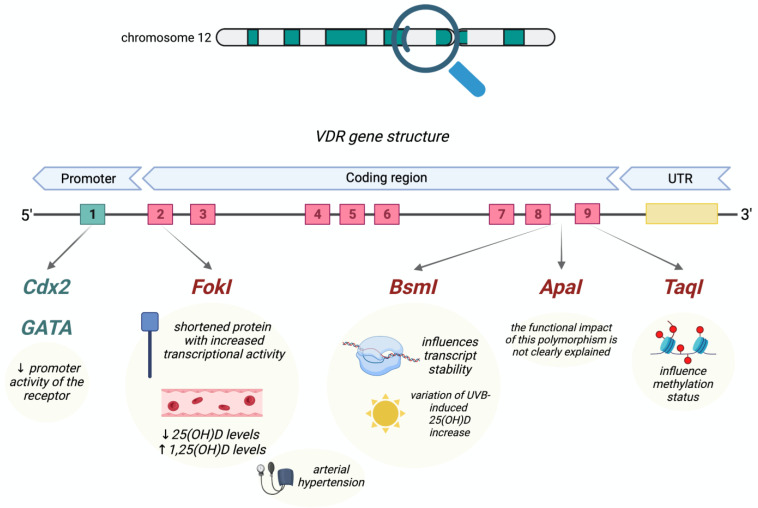
The main polymorphisms characterizing a vitamin D receptor. Each one is pictured below its corresponding exon (indicated by progressive numbers from 1 to 9) matched via black arrows, and with its known clinical implications. VDR: vitamin D receptor, 25(OH)D: 25-hydroxylated vitamin D, 1,25(OH)D: 1,25-hydroxylated vitamin D.

**Figure 3 genes-14-01691-f003:**
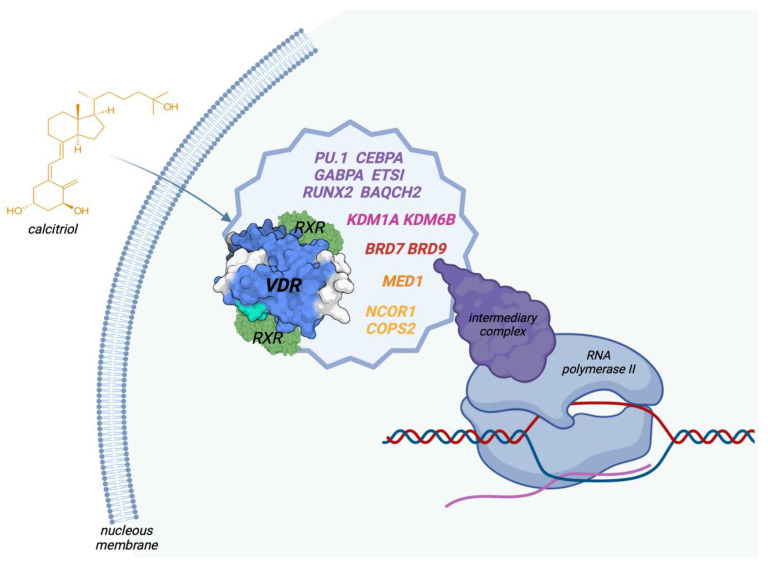
Describing the multi-complex of the VDR, comprising the receptor, co-receptors, co-repressors, co-activators, and pioneer factors that finally exert genomic effects on gene transcription. VDR: vitamin D receptor, RXR: retinoid acid receptor, *PU.1*, *CEBPα*, *GABPα*, *ETS1*, *RUNX2*, *BACH2:* pioneer factors, KDM1A, KDM6B: chromatin modifiers, BRD7, BRD9: chromatin remodelers, MED1: co-activator, NCOR2, COPS2: co-repressors.

## Data Availability

All data presented are included in the manuscript.

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
