# Peer review of "Vitamin D: An Overview of Gene Regulation, Ranging from Metabolism to Genomic Effects"

_genes, 2023, doi:10.3390/genes14091691_

Round 1

Reviewer 1 Report

Major comments:

1. The title of the manuscript is misleading and should be adapted. Only lines 287-311 describe what a reader may expect from the present title (gene regulation), but this is insufficient. Moreover, there is no indication of any "interference.

2. For a review there is an insufficient number of figures, there should be at least 4. Moreover, Fig. 1 is very superficial and not very accurate, it needs to be revised. Please refer to recent expert literature.

3. Please retain from abbreviating the term "vitamin D" but write it out. In parallel, please be more accurate when using the term, whether vitamin D3, 25(OH)D3 or 1,25(OH)2D3 is meant.

4. The authors should study the vitamin D literature more intense. For example, due to last year's 100 years vitamin D "birthday" there were a number of good review articles published. In fact, since most things about vitamin D are already said, what is the new aspect of this manuscript. Please highlight new insight.

5. It is not clear why the authors emphasize the polymorphisms (better use consistently the term "SNP") of VDR and related genes. Anyway, please be consistent in nomenclature and make clear why you consider this aspect so important while others (e.g. mechanisms of gene regulation) are largely neglected.

Minor comments:

1. Gene name abbreviations should be in italic.

2. All abbreviations should be defined at first time use and then consistently applied. This applies also to gene names.

acceptable

Reviewer 2 Report

First of all, I’d like to congratulate the authors on the manuscript with performed work and writing an excellent review on the topic of Vitamin D genetics. The manuscript is very well-written and presents an interesting data on vit D metabolism which summarizes and structures the data of multiple studies on the subject. Nevertheless, I have a number of suggestions for the manuscript, which in no way reduce its quality and are offered at the discretion of the authors.

1.       According to HGVS recommendations for the description of sequence variants: 2016 update [http://onlinelibrary.wiley.com/doi/10.1002/humu.22981/pdf], in some disciplines the term “polymorphism” is used both to indicate “a non-disease-causing change” or “a change found at a frequency of 1% or higher in a population”. To prevent this confusion, it is not recommended to use the terms polymorphism (including SNP or Single Nucleotide Polymorphism) but neutral terms like “sequence variant”, “alteration”, “allelic variant”, and SNV or Single Nucleotide Variation.

2.       The human gene abbreviations should be written in italics

3.       Lines 37-39, “Interestingly, VDR is actually expressed in almost every type of human cell, and it has been found to modulate the transcription of about 3% of human genes []”. – reference is required.

4.       Every section has an explanation of the VDR abbreviation, but none has PTH decoding.

5.       There is a lot of information in the text on CYP genes, but no mention on cytochrome P450 superfamily, description of which should be added

6.       Chapter 2, the section details the production of the vit D in the kidneys, but no information on extrarenal production.

7.       Line 128, “The main steps of vitamin D metabolism and its genomic and NON-genomic effects are depicted in Figure 1”.

8.       Chapter 4, when describing genes, involved in vit D metabolism, simple enumeration of rs-numbers of loci is non-informative. Most readers would be more interested in detailed information on each locus: which genotype is associated with which effect on vitamin D levels. Section 4.6 provides a more detailed description of these VDR gene variants, but again does not specify the mechanism by which specific genotype decrease or increase serum vitamin D levels.

9.       Section 4.6, it is better to come to the uniformity of the indication of alleles – it should be reported in the Forward stand orientation (according to GRCh38/hg19), like in NCBI database.

10.   Line 228, for Bsmi variant, the reference and alternative alleles are rearranged, there should be G>A nucleotide substitution.

11.   Line 284, of cell growth aNd differentiation

12.   Duplication of 13 and 15 references (B.W.Hollis) and 54 and 61 references (E.A.Hibler et al.).

Reviewer 3 Report

Paper is good, but need some language revision

The topic is unique and worthy of researching, as this review aimed to provide an overview of the genetic aspects related to vitamin D and VDR, emphasizing the gene regulation of vitamin D and the genetic modulation of VDR target genes. In addition, to examine the pathogenic role of the most known VDR polymorphisms and report a brief summary of the rare genetic diseases linked to vitamin D metabolism. The deduced conclusions based on the research methods/cases are enough and tenable. Regarding, the progress that had been made compared with the current research results; this Review highlights and provides an overview of the genetic aspects of vitamin D and VDR, emphasizing the gene regulation of vitamin D and the genetic modulation of VDR target genes. In addition, briefly summarizes the rare genetic disease linked to vitamin D metabolism.

Strengths and weaknesses

The abstract is informative and reflect the body of the paper. The introduction provides sufficient background information for readers in the immediate field to understand the problem/hypotheses. The text is well arranged and the logic, the related concepts introduced clearly and the readability is sufficient. The discussion and theoretical analysis in this article are good. The reference section is informative and accurate.

Suggestions for improvement

Paper is good, but need some language revision.

Round 2

Reviewer 1 Report

The manuscript is insufficiently revised, please read my comments carefully and react accordingly

OK

Author Response

We read again the previous revisions. Despite confirming what we wrote in the previous response we further enrich the manuscript in section 5.1. More genetic aspects about the VDR signaling have been added, expanding the genomic landscape of this topic. Moreover, we created one more figure, making the manuscript extremely balanced in terms of text/figure ratio. 

Thank you again for the precious comment which push up to improve the paper